# Development of an Anthropomorphic Heterogeneous Female Pelvic Phantom and Its Comparison with a Homogeneous Phantom in Advance Radiation Therapy: Dosimetry Analysis

**DOI:** 10.3390/medsci11030059

**Published:** 2023-09-11

**Authors:** Neha Yadav, Manisha Singh, Surendra P. Mishra, Shahnawaz Ansari

**Affiliations:** 1Department of Applied Physics, Amity School of Engineering & Technology, Amity University Madhya Pradesh, Maharajpura Dang, Gwalior 474005, India; msingh@gwa.amity.edu; 2Department of Medical Physics, Apollo Hospitals Bilaspur, Bilaspur 495006, India; ansarisnz05@gmail.com; 3Department of Radiation Oncology, Dr. Ram Manohar Lohia Institute of Medical Sciences, Lucknow 226010, India; mishrasp05@gmail.com

**Keywords:** homogeneous phantom, anthropomorphic heterogeneous phantom, radiation therapy, patient dosimetry, treatment planning

## Abstract

Background: Accurate dosimetry is crucial in radiotherapy to ensure optimal radiation dose delivery to the tumor while sparing healthy tissues. Traditional dosimetry techniques using homogeneous phantoms may not accurately represent the complex anatomical variations in cervical cancer patients, highlighting the need to compare dosimetry results obtained from different phantom models. Purpose: The aim of this study is to design and evaluate an anthropomorphic heterogeneous female pelvic (AHFP) phantom for radiotherapy quality assurance in cervical cancer treatment. Materials and method: Thirty RapidArc plans designed for cervical cancer patients were exported to both the RW3 homogeneous phantom and the anthropomorphic heterogeneous pelvic phantom. Dose calculations were performed using the anisotropic analytic algorithm (AAA), and the plans were delivered using a linear accelerator (LA). Dose measurements were obtained using a 0.6 cc ion chamber. The percentage (%) variation between planned and measured doses was calculated and analyzed. Additionally, relative dosimetry was performed for various target locations using RapidArc and IMRT treatment techniques. The AHFP phantom demonstrated excellent agreement between measured and expected dose distributions, making it a reliable quality assurance tool in radiotherapy. Results: The results reveal that the percentage variation between planned and measured doses for all RapidArc quality assurance (QA) plans using the AHFP phantom is 10.67% (maximum value), 2.31% (minimum value), and 6.89% (average value), with a standard deviation (SD) of 2.565 (t = 3.21604, *p* = 0.001063). Also, for the percentage of variation between homogeneous and AHFP phantoms, the t-value is −11.17016 and the *p*-value is <0.00001. The result is thus significant at *p* < 0.05. We can see that the outcomes differ significantly due to the influence of heterogeneous media. Also, the average gamma values in RapidArc plans are 0.29, 0.32, and 0.35 (g ≤ 1) and IMRT plans are 0.45, 0.44, and 0.42 (g ≤ 1) for targets 1, 2, and 3, respectively. Conclusion: The AHFP phantom results show more dose variability than homogenous phantom outcomes. Also, the AHFP phantom was found to be suitable for QA evaluation.

## 1. Introduction

Cervical cancer is a significant global health concern and a leading cause of cancer-related deaths among women in many developing countries. Treatment modalities for cervical cancer depend on the stage and extent of the disease. They may include surgery, chemotherapy, immunotherapy, and radiation therapy. Radiotherapy plays a crucial role in the management of cervical cancer, especially in its early and locally advanced stages. It can be used as a primary treatment, combined with surgery or chemotherapy, or as a postoperative adjuvant therapy to prevent disease recurrence. The role of radiotherapy in cervical cancer treatment is to deliver a targeted dose of ionizing radiation to the tumor, destroying cancerous cells and shrinking tumors while sparing adjacent normal tissues. It is an effective and non-invasive treatment option that can achieve high cure rates and preserve fertility in early-stage cases, making it an indispensable component of comprehensive cervical cancer management [1].

The paramount role of physics in radiation therapy is to continually enhance the precision and accuracy of delivering the radiation dose to the target volume. The historical evolution of radiation therapy witnessed a paradigm shift from a two-dimensional (2D) approach prevailing from the 1950s to the late 1980s. In this era, radiation plans were manually designed, and a single radiation beam was delivered from one to four directions using specialized shielding blocks for beam collimation [2,3]. The advent of advanced imaging technologies, such as ultrasound (US), computed tomography (CT), and magnetic resonance imaging (MRI), led to a transformative transition to three-dimensional conformal therapy (3DCT). This revolutionary approach enables precise treatment field shaping to the target volume, ensuring uniform-intensity delivery to the tumor while sparing surrounding healthy tissues. The innovative multileaf collimator (MLC) system replaced traditional shielding blocks, which significantly improved treatment accuracy [4,5]. In the 1990s and early 2000s, intensity-modulated radiation therapy (IMRT) emerged as a major advancement in cancer treatment. It provided more precise and conformal dose distributions compared to the traditional 3D conformal radiation therapy (3DCRT) technique [6,7]. This led to further developments, such as intensity-modulated arc therapy (IMAT), which used dynamic manipulation of the multileaf collimator (MLC) while rotating the treatment machine’s gantry in an arc. IMAT then paved the way for volumetric modulated arc therapy (VMAT) in 2007, which incorporated variable gantry rotation speeds and dose rates, making treatments even more accurate and efficient. These advanced techniques have revolutionized modern radiation therapy, improving treatment outcomes and quality of life for cancer patients while reducing radiation exposure to healthy tissues [8,9,10,11].

Also, these high-end radiation therapy techniques require accurate pretreatment and patient-specific quality assurance (PSQA) prior to the start of patient treatment [12,13]. Two important factors must be considered in the evaluation of any radiotherapy plans or treatment procedures: a realistic environment that mimics how radiation interacts with real biological tissue, and a precise pretreatment plan verification system [14]. Phantoms, which have been in use since the inspection of radiotherapy, are substitutes that conform to the real-body scenario. Although the majority of the human body consists of water, physical phantoms that are made of water or solid water-equivalent materials have mostly been used for PSQA [15,16]. These phantoms were used because of their cost effectiveness, universal availability, uniform density of 1 g/cc, and simpler designs. However, we know that in addition to water, the human body consists of bones, soft tissues, air cavities, etc., of varying densities. So, there is a need to develop an anthropomorphic heterogeneous pelvic phantom that should exactly represent the actual human body. Previous studies must be consulted to ensure that the phantom would be constructed to be realistic in size and shape [17,18,19]. The phantom would also be suitable for the assessment of accurate delivery of treatment doses, and improve dosimetry in clinical fields. By using this phantom, uncertainties during patient positioning and dosimetry can be reduced, making it an important tool for practical tests in India.

## 2. Materials and Methods

### 2.1. Phantom Design

To design an anthropomorphic heterogeneous female pelvic (AHFP) phantom, the average pelvic dimensions of 50 adult female patients were utilized, as shown in Figure 1A. To accurately replicate the radiological characteristics of the involved tissues, a combination of materials was chosen, including paraffin wax, water, gauze (cotton), polyvinyl chloride (PVC), and polymerized siloxanes, as seen in Table 1. For the uterine part, we mixed 150 g of polymerized siloxanes with 50 g of regular wax. This mixture helped us make a structure that feels like a real uterus. The rectum simulation involved a combination of materials: a PVC hollow pipe, paraffin wax, and a thin gauze piece. The foundation was the 14 cm long, 1.5 cm diameter, and 1 mm thick PVC pipe, housing 10 g of paraffin wax for soft tissue emulation. A 10 cm long, 1 mm thick gauze piece was inserted within the pipe to simulate fecal matter, making the rectal part more realistic. For the bladder, we used a balloon and filled it with 220 mL of water. This made the balloon act like a real bladder filled with urine. We combined paraffin wax with sodium chloride (salt) to replicate the characteristics of fat and skin.

### 2.2. Fabrication of Phantom

Initially, a female pelvic dummy was meticulously crafted employing thermoplastic sheets and cloth tape. Subsequently, internal organ models were precisely situated to mirror the density of human pelvic bone, and their placement was secured through the application of gypsum bandages. These internal organs, along with the pelvic bones, were harmoniously integrated into the pelvic dummy, ensuring anatomical alignment. We then poured liquid paraffin wax for surface molding, subsequently allowing it to cool and stabilize. A cavity was prepared approximately at the uterus area in the phantom, and for that purpose, a 0.60 cc ion chamber (PTW, Freiburg, Germany) was kept at the same position to make sure that the cavity’s dimensions were equal to the ion chamber. Additionally, three reference points were created using fiducial lead markers placed on two bilateral points and one anterior point on the phantom’s surface within the same cross-sectional plane.

The phantom’s physical measurements are 25.5 cm in terms of anterior–posterior separation, 32.7 cm in terms of lateral separation, and around 31.8 cm in the vertical dimension, with the extent being from the lower abdomen to the upper thigh region. The phantom weighs approximately 14.8 kg.

### 2.3. Comparison of the Hounsfield Units and the Relative Electron Densities of the Organs

To determine how accurately the finished phantom product represents a real patient, the AHFP phantom was scanned with a CT scanner (Toshiba Alexion 16 multi-slice CT scanner) at 120 kVp and 250 mAs with a slice thickness of 2 mm. The CT images were transferred to the Eclipse treatment planning system (version 11.0.31) (Varian Medical Systems, Palo Alto, CA, USA). The CT images of the phantom were compared to CT images of randomly selected cervical cancer patients with similar scanning parameters (120 kVp, 250 mAs, and 2 mm slice thickness), which are shown in Figure 2A,B.

Table 2 shows the mean and standard deviation of the CT number in Hounsfield units (HU) for patient and phantom CT images. The relative electron density of each of the materials was calculated by the given Formulas in (1) and (2) (Thomas, 2014) [20].
*P_e_* = *HU*/1000 + 1 *HU* < 100(1)
*P_e_* = *HU*/1950 + 1 *HU* ≥ 100(2)
where *p_e_* is the relative electron density of the material.

### 2.4. Anatomical and Measuring Point Identification

The anatomical structures, including the external body, skeletal or bone structure, bladder, rectum, uterus, femoral heads, and pelvic bone, were meticulously delineated using the Eclipse contouring station (Version 11.0.31). The delineated structures and contours were then exported to the Eclipse planning system, allowing for 3D visualization of each structure. To facilitate accurate dosimetry measurements, an ionization chamber was strategically positioned at the clinic’s areas of interest, particularly near critical structures. This allowed for precise monitoring and recording of the absorbed dose at these specific locations during the course of the treatment.

### 2.5. Pretreatment Plan Verification

#### 2.5.1. Patient-Specific Absolute Dosimetry

Two kinds of phantoms were chosen for the patient-specific absolute dosimetry of the completed RapidArc treatment plans. The first one was a homogeneous “water-equivalent RW3 solid phantom” (PTW Freiburg, Freiburg, Germany), as shown in Figure 1B, each slab of which was made of polystyrene with the effective atomic number 5.74. The second phantom was the AHFP phantom, as shown in Figure 1A. The density of the internal organs of this AHFP phantom was equivalent to that of the human pelvis. The CT scanning of the phantoms was conducted on a Toshiba Alexion 16 multi-slice CT scanner, with a slice thickness of 2 mm for planning purposes. The CT images were imported into the Eclipse (version 11.0.31) TPS (Varian Medical Systems, Palo Alto, CA, USA), and RapidArc plans already conducted for patient treatment were exported into both phantoms, which can be seen in Figure 3A,B.

Thirty cervical cancer patients who underwent RapidArc therapy, ranging in age from 37 to 70 years (average 53.5 years), were selected randomly for the study. Dual arcs were used for all the RapidArc plans since dual arcs can improve PTV coverage, enhance the modulation factor during optimization, and spare the OARs compared to single arcs. The first arc was a clockwise rotation with a gantry angle of 181° to 179° and a collimation angle of 30°. The second arc had a collimation angle of 330° and an anticlockwise rotation with gantry angles of 179° to 181°. All the selected plans were performed with a 6 MV photon beam, and field arrangement was conducted in such a way that all fields were coplanar with a couch angle of 0°. A dose volume optimizer (DVO) was used for plan optimization, and an anisotropic analytical algorithm (AAA) (version 11.30.1) with a grid size of 0.25 cm was used for dose calculation. All the plans were delivered, and the dose for each plan was measured using a PTW UNIDOSE electrometer connected with a 0.6 cc ionization chamber (IBA Dosimetry Germany), which was fixed in phantoms.

The percentage (%) variation between the measured dose of the linear accelerator and the planned dose of the TPS was calculated by the following formula:Percentage of variation = (measured dose of linac − TPS planned dose)/TPS planned dose × 100

The planned doses of the TPS and the measured doses from the machines of the homogeneous phantom and the AHFP phantom are compared and represented in Table 3 and Table 4, respectively.

#### 2.5.2. Relative Dosimetry

To evaluate the effectiveness of the AHFP phantom as a quality assurance (QA) tool, planning target volumes (PTVs) were generated for the phantom. To assess the dose received by healthy organs during radiation therapy, organs at risk (OARs), like the bladder and rectum, were also considered. Both RapidArc and intensity-modulated radiation therapy (IMRT) plans were created on a treatment planning system (TPS), and the anisotropic analytical algorithm (AAA) (version 11.0.31) was used to calculate the dose. The 2D fluence generated by the TPS on the electronic portal imaging device (EPID) was sent to the linear accelerator (linac) for further analysis, which is represented in Figure 4. Most modern linacs are equipped with flat-panel detectors based on amorphous silicon (aS1000 model) for megavoltage imaging. Various methods have been developed to utilize EPIDs in IMRT/RapidArc patient-specific quality assurance (PSQA). All measurements were carried out using an EPID detector calibrated for a 100 cm source-to-imager distance (SID). Data collection was performed with the same gantry and collimator positions specified in the treatment plan.

The imaging system software was employed to compare and analyze the 2D fluence imaging obtained from the treatment planning system (TPS). For plan evaluation, pixel-based passing criteria were utilized. A pass condition was set at an average gamma value (g) of ≤1, indicating successful plan agreement, while a failure condition was defined as g > 1, indicating discrepancies. In our assessment, we used specific tolerance levels as acceptance standards. These included a distance-to-agreement (DTA) of 3 mm, representing the maximum allowed spatial difference between the measured and expected dose distributions. Additionally, a dose difference (DD) of 3% was considered, signifying the permissible variation between the measured and expected doses.

### 2.6. Statistical Analysis

The statistical analysis in this study involved a paired two-tailed Student’s *t*-test using SPSS^®^ v.13.0 (SPSS Inc., Chicago, IL, USA) to compare the differences between the homogeneous RW3 phantom and the AHFP phantom. A significance level of *p* < 0.05 was considered as statistically significant to determine the presence of significant differences.

## 3. Results

Overall, there is good agreement between the measured CT number (HU) and relative electron density (RED) of the AHFP phantom and the patient groups. Table 2 displays the findings of the comparison between measured CT numbers from a sample of patients from our institution who were selected at random and the CT numbers of the phantom. Hence, it was observed that the AHFP fabricated for this study matched both the qualitative and quantitative aspects of the CT evaluation.

In the case of the homogeneous phantom, the mean percentage variations between planned and measured doses of all rapid arc QA plans were 1.4299, and the standard deviation was 0.768 (t = 0.00508, ρ = 0.497982). The result is not significant at *p* < 0.05, as shown in Table 3. For the AHFP phantom, the mean percentage variations between planned and measured doses of all rapid arc QA plans were 6.890, and the standard deviation was 2.565 (t = 3.21604, ρ = 0.001063 < 0.05). The outcome is significant, as shown in Table 3. The comparative study of the percentage of variation between the homogeneous slab phantom and AHFP phantom is given in Table 4. The t-value is −11.17016 and the *p*-value is <0.00001. The result is significant at *p* < 0.05, and their graphical representation is shown in Figure 5.

The results obtained from relative dosimetry are tabulated in Table 5. The average gamma values of the RapidArc plans are 0.29, 0.32, and 0.35 (g ≤ 1), and these values for the IMRT plans are 0.45, 0.44, and 0.42 (g ≤ 1) for targets 1, 2, and 3, respectively.

## 4. Discussion

The results of our research, as presented in Table 2, demonstrate a close similarity between the Hounsfield unit (HU) and relative electron density (RED) values of the locally manufactured AHFP phantom and those of a human female pelvis. This finding aligns with previous research that emphasizes the importance of phantom design for dose measurement accuracy Johns & Cunningham [21]. Our study emphasizes the significance of tissue-equivalent phantoms, such as the AHFP, in achieving clinically relevant and precise dose measurements, as discussed by Almond et al. [22]. Numerous studies have contributed valuable data on the Hounsfield units and relative electron densities of human tissues, supporting the consistency and accuracy of these measurements. Winslow et al. [23] determined the Hounsfield units for human muscles, and established the soft tissue equivalent range as −55 to −155, with the bone tissue equivalent range being 660. This aligns with the research by Trujillo-Bastidas et al. [24] and Kanematsu [25], which reported relative electron densities for adipose, muscle, and bone tissues as 0.97, 1.05, and 1.4 and 0.96, 1.06, and 1.12, respectively. Similarly, the research conducted by Shrotriya et al. [26] and S. Singh et al. [27] revealed relative electron density values for bladder, rectum, fat, and bone tissues that closely align with our study’s findings (1.015, 1.069, 0.909, and 1.628, respectively). Shrotriya et al. (2018) reported relative electron densities of 1.31, 1.025, 0.91, and 1.6, respectively, while S. Singh et al. (2020) observed values of 1.04, 1.05, 0.89, and 1.63 for the same tissues. These consistent results across studies contribute to the overall understanding and validation of relative electron density values for different tissues, adding to the reliability of dosimetry calculations in radiation therapy planning.

A variety of techniques have been created to compare sets of planned and measured radiation dose distributions in radiotherapy dosimetry. Here, we compare the homogeneous phantom’s and the AHFP phantom’s measured dose of the linac (Clinac iX medical linear accelerator, Varian Medical System, Palo Alto, CA, USA) and the planned dose of the Eclipse planning system (Version 11.0.31) (Varian Medical System, Palo Alto, CA, USA). In the case of the homogenous phantom, there is a less than 3% difference in the percentage between planned and measured doses, with a standard deviation of 0.7682 (t = 0.00508, *p* = 0.497982. The result is not significant at *p* < 0.05. The deviations of the planned and measured values of the dose of the AHFP phantom were found to be 10.67% (maximum value), 2.31% (minimum value), and 6.89% (average value), with a standard deviation of 2.565 (t = 3.21604, *p* = 0.001063). The result is significant at *p* < 0.05. Also, for the percentage of variation between homogeneous phantoms and AHFP phantoms, the t-value is −11.17016 and the *p*-value is <0.00001. The result is therefore significant at *p* < 0.05. In Figure 5, we can see that the outcomes differ significantly due to the influence of heterogeneous media. The observed deviations in dose measurements between planned and measured values in the AHFP phantom could have potential clinical implications. Higher deviations could lead to suboptimal treatment delivery, affecting treatment outcomes and patient safety. Similar concerns have been raised in previous research, indicating the importance of minimizing dose calculation errors to improve treatment quality and patient safety (ICRU Report 50 [28]).

The human body is made up of various densities, such as fat, bones, air cavities, and tissue. The amount of radiation dose deposited at the interface of two mediums varies significantly due to the difference in electron densities between the two media. Because bones have a larger density than soft tissue, they produce more secondary electrons [29,30]. As a result, the dosage at the bone–soft tissue interface is higher. A similar phenomenon occurs at the interface of all two metals with different densities. Heterogeneity is one of the hardest problems that dose calculation algorithms must solve. The TPS currently uses newer and more precise algorithms that, like AAA, apply the heterogeneity adjustment factor when calculating dose [31,32]. The patient-specific absolute dosimetry should be carried out using a heterogeneous phantom that mimics the density of the human body to confirm the correctness of the dose computed by these algorithms in the instance of each patient. O. Gurjar et al. conducted a study on radiation dosimetry for a contemporary radiotherapy approach, employing a real tissue phantom [33]. With IMRT (head phantom) and IMRT (tissue phantom), the mean percentage deviation between planned and measured doses was found to be 2.36 (SD: 0.77) and 3.31 (SD: 0.78), respectively. Although the percentage variation in the case of the head phantom was within the tolerance limit (3%), it was nonetheless larger than the outcomes obtained utilizing phantoms that were readily available in the marketplace [34,35]. And the majority of tissue phantom cases had percentage variations that exceeded the tolerance level. Chen et al. [36] and Lee et al. [37] conducted dosimetric validation and accuracy assessment of an in-house-developed anthropomorphic heterogeneous female pelvic phantom. Their findings showcased the suitability of the phantom for radiotherapy quality assurance, confirming its ability to accurately simulate patient anatomy and tissue heterogeneity. This study emphasizes the importance of utilizing reliable phantoms to ensure precise dose delivery and patient safety during treatment.

Collectively, these studies highlight the importance of utilizing anthropomorphic heterogeneous female pelvic phantoms for accurate dosimetric verification in radiotherapy. The use of such phantoms improves confidence in treatment planning processes and ensures optimal dose delivery to target volumes while sparing healthy tissues. Further research and validation of these phantoms on a larger scale will likely enhance their clinical applicability and contribute to improved patient outcomes in radiation therapy.

In Table 5, we present a comprehensive assessment of all planned target locations on the AHFP phantom, utilizing both RapidArc and IMRT treatment techniques. The table provides essential parameters, including area gamma, maximum dose difference, and average dose difference, for each target location. The results highlight the remarkable agreement between the measured and expected dose distributions, as indicated by the area gamma values ranging from 97.9% to 99.8% across all target locations. This excellent level of concurrence underscores the AHFP phantom’s ability to faithfully replicate radiation interactions and accurately deliver doses to the intended target volumes. Examining the maximum dose difference, we observe variations ranging from 18.7% to 39.4% for the different target locations. Additionally, the average dose difference ranges from 1.1% to 2.5%. These values, although demonstrating some variability, remain well within acceptable tolerance limits for relative dosimetric purposes. Our research findings are in line with the study conducted by Smith et al. [38]. By utilizing the gamma index approach, both studies assessed the agreement between calculated and measured dose distributions, which provided crucial insights into the accuracy of treatment planning. The study by Smith et al. reinforces the significance of gamma index analysis as a valuable and comprehensive tool for dosimetric evaluation, further validating its importance in radiation therapy quality assurance and treatment optimization.

Overall, these findings underscore the AHFP phantom’s effectiveness as a reliable quality assurance (QA) tool in radiotherapy. Its capability to mimic human anatomy and accurately simulate radiation dose delivery allows it to play a crucial role in verifying treatment plans and ensuring precise dose administration to target areas. The AHFP phantom’s performance, as evidenced by these results, reinforces its significance in advancing the safety and efficacy of radiotherapy procedures.

## 5. Conclusions

This study highlights the importance of precise dose delivery in radiotherapy for cervical cancer treatment. The anthropomorphic heterogeneous female pelvic (AHFP) phantom successfully replicates the radiological characteristics of human tissues, providing a realistic environment for accurate dose measurements. The comparison between the AHFP phantom and patient CT images demonstrates close similarity, confirming the phantom’s suitability for patient-specific quality assurance.

The AHFP phantom’s measured dose variations using RapidArc plans indicate significant discrepancies compared to homogeneous phantoms, emphasizing the impact of heterogeneous media on dose calculations. However, the AHFP phantom’s performance remains within acceptable limits for relative dosimetric purposes. Furthermore, the AHFP phantom proves effective in plan verification, with high agreement between measured and expected dose distributions. The phantom’s ability to simulate human anatomy and accurately deliver doses to target volumes reinforces its role as a reliable QA tool in radiotherapy.

Overall, the AHFP phantom’s development and evaluation contribute to advancing the precision and efficacy of radiotherapy for cervical cancer, ultimately improving patient outcomes, and ensuring safer and more effective treatment strategies.

## Figures and Tables

**Figure 1 medsci-11-00059-f001:**
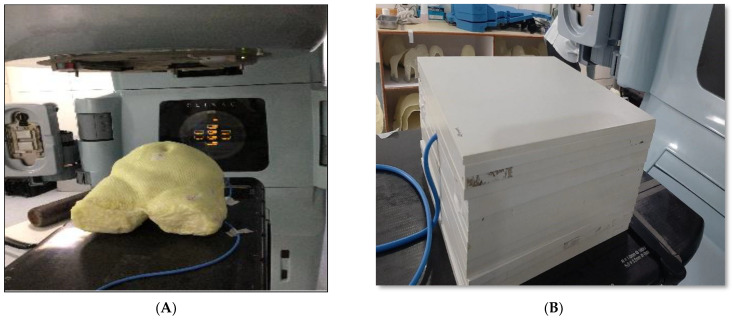
(**A**) Setup of the anthropomorphic heterogeneous female pelvic (AHFP) phantom on a linear accelerator (Varian Medical Systems, Palo Alto, CA, USA) for dosimetry study and (**B**) setup of the homogeneous RW3 phantom.

**Figure 2 medsci-11-00059-f002:**
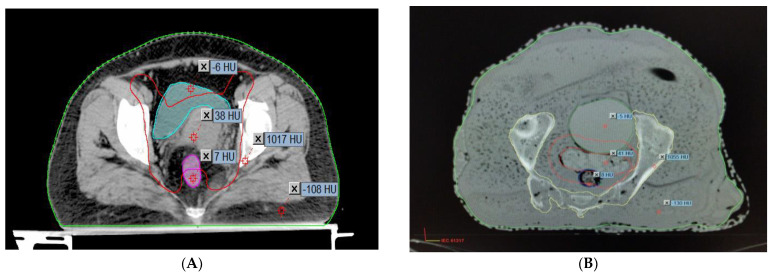
HU value representation on CT images of (**A**) a real female patient and (**B**) the AHFP phantom.

**Figure 3 medsci-11-00059-f003:**
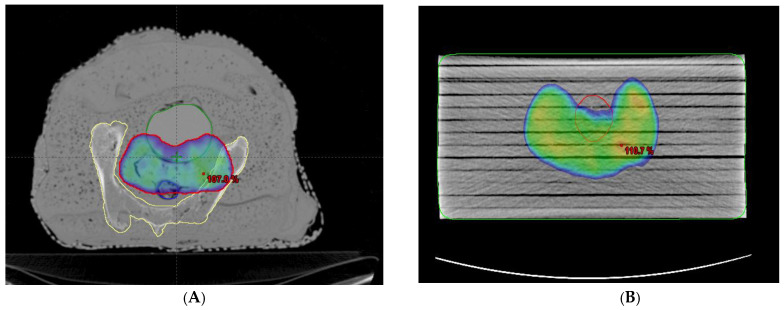
(**A**,**B**) RapidArc plan representation on the AHFP phantom and homogeneous (RW3) phantom with dose coverage of 95% of prescribed.

**Figure 4 medsci-11-00059-f004:**
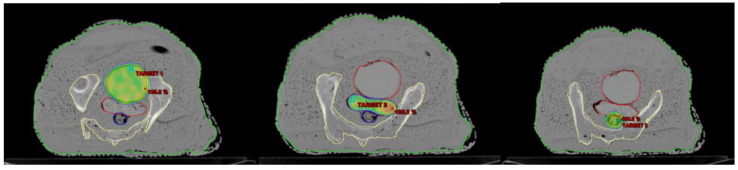
Displaying three different targets on the AHFP Phantom for dose verification.

**Figure 5 medsci-11-00059-f005:**
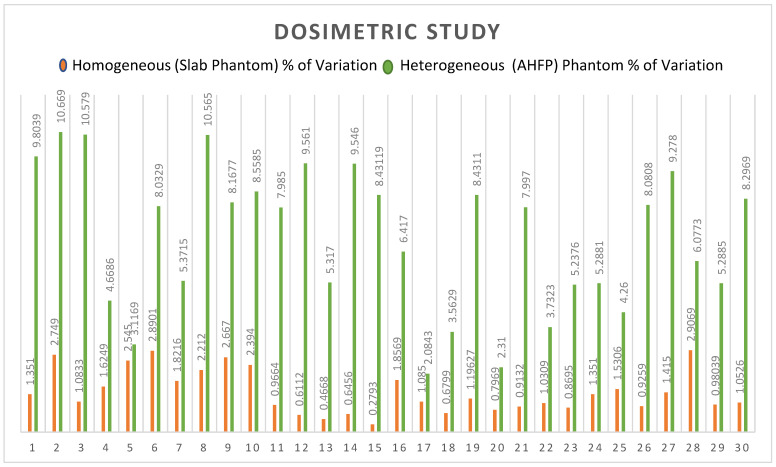
Graphical representation of percentage of variation of the homogeneous slab phantom and the AHFP phantom.

**Table 1 medsci-11-00059-t001:** Composition of tissue-equivalent materials for AHFP phantom development.

AHFP Phantom	Composition of Materials
Fat and skin	Paraffin wax (13 kg) and NaCl (50 g)
Muscles	Silicon sealant (100 g) and paraffin wax (200 g)
Bladder	Balloon filled with 220 mL of water
Rectum	Polyvinyl chloride, gauze, and paraffin wax (10 g)
Uterus	Polymerized siloxanes (150 g) and wax (50 g)
Bone	Pelvic bone: human equivalent

**Table 2 medsci-11-00059-t002:** Hounsfield unit (HU) and relative electron density (RED) measurements of the developed AHFP phantom and real patients.

S.N.	Pelvic Organs	Actual Female Patients		AHFP Phantom	
		HU ± SD	RED	HU ± SD	RED
1	Uterus	45 ± 20	1.031	50 ± 21	1.07
2	Bladder	12 ± 6	1.02	−4.0 ± 17	1.015
3	Rectum	42 ± 17	1.040	43 ± 26	1.069
4	Muscle	70 ± 12	1.08	72 ± 33	1.105
5	Fat	−120 ± 8	0.955	−170 ± 79	0.909
6	Bone	965 ± 110	1.489	947 ± 277	1.628

**Table 3 medsci-11-00059-t003:** Percentage variations between planned dose of the TPS and measured dose of the linear accelerator using the RW3 phantom and AHFP phantom.

	Homogeneous Phantom (RW3 Phantom)			Heterogeneous Phantom (AHFP)		
Sr. No.	Planned Dose of TPS (cGy)	Measured Dose of LA (cGy)	% of Variation	Planned Dose of TPS (cGy)	Measured Dose of LA (cGy)	% of Variation
1	199.01	196.32	−1.352	204	184	−9.804
2	200.4	194.89	−2.749	210.6	188.13	−10.669
3	192	189.92	−1.083	214	191.36	−10.579
4	230.16	233.9	1.625	203.27	193.78	−4.669
5	200.38	205.48	2.545	194.1	188.05	−3.117
6	216.25	210	−2.890	212.5	195.43	−8.033
7	185	181.63	−1.822	205.9	194.84	−5.372
8	220.13	225	2.212	192.6	172.25	−10.566
9	205.09	199.62	−2.667	226.5	208	−8.168
10	205.09	210	2.394	210.9	192.85	−8.558
11	172.8	171.13	−0.966	200	184.03	−7.985
12	196.3	197.5	0.611	210	189.92	−9.562
13	192.8	193.7	0.467	230	217.77	−5.317
14	196.7	195.43	−0.646	216	195.38	−9.546
15	200.5	199.94	−0.279	218	199.62	−8.431
16	187.4	183.92	−1.857	199	186.23	−6.417
17	202.7	200.5	−1.085	220.7	225.3	2.084
18	205.9	204.5	−0.680	210.5	218	3.563
19	225.7	228.4	1.196	218	199.62	−8.431
20	197	195.43	−0.797	200	195.38	−2.31
21	219	221	0.913	172.8	158.98	−7.997
22	194	192	−1.031	191.3	198.44	3.732
23	230	228	−0.869	197.8	187.44	−5.238
24	185	187.5	1.351	189.48	199.5	5.288
25	196	199	1.531	223	213.5	−4.260
26	216	218	0.926	198	182	−8.081
27	212	215	1.415	194	176	−9.278
28	172	177	2.907	181	192	6.077
29	204	206	0.980	208	197	−5.288
30	190	188	−1.053	229	210	−8.297

**Table 4 medsci-11-00059-t004:** Summary of statistical data.

Sr. No.	Statistical Parameters	Homogeneous (RW3) Phantom	AHFP Phantom
1	N	30	30
2	∑X	42.898	206.715
3	Mean	1.429	6.890
4	∑X^2^	78.454	1615.167
5	SD	0.768	2.565
6	t-value	0.005	3.216
7	ρ	0.498	0.001

Note: N = number of patients; SD = standard deviation; ρ = significance value.

**Table 5 medsci-11-00059-t005:** Comprehensive evaluation of planning target locations on the AHFP phantom.

Target Location	Treatment Technique	Area Gamma	Maximum Dose Difference	Average Dose Difference
Target 1	RapidArc	99.8%	18.7%	1.1%
Target 2	RapidArc	97.9%	25.6%	1.2%
Target 3	RapidArc	98.8%	20.2%	1.3%
Target 1	IMRT	99.5%	26.5%	1.95%
Target 2	IMRT	99.6%	39.4.0%	2.5%
Target 3	IMRT	98.6%	20.2%	2.0%

## Data Availability

The data presented in this study are available on request from the corresponding author. The data are not available due to privacy restrictions.

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
