# Peer review of "Development of an Anthropomorphic Heterogeneous Female Pelvic Phantom and Its Comparison with a Homogeneous Phantom in Advance Radiation Therapy: Dosimetry Analysis"

_medsci, 2023, doi:10.3390/medsci11030059_

Round 1

Reviewer 1 Report

Dear authors, I have read the manuscript.

The objectives are clear and all parts are connected quite well.

I have four comments.

1 Improve your English

2 Review the approximations of the tables (too many decimal places)

3 Add some statistical comparisons

4 Verify the radiological tissue-equivalence (see for example 10.1088/1361-6463/ab08d0)

Good Manuscript

Author Response

Thank you for giving me the opportunity to submit a revised draft of my manuscript titled Development of anthropomorphic heterogeneous pelvic phantom and its comparison with homogeneous phantom in advance radiation therapy: Dosimetry Analysis to MDPI Journal. We are complying your valuables suggestions and comments in the revised manuscript.

Reviewer 2 Report

The article presents an interesting implementation in medical physics of common everyday life techniques, such as the FDM 3d printing.

Nevertheless, the authors need to be more specific on the exact materials used to produce this heterogeneous phantom. Authors need to provide more information on the exact proportion of materials utilized (exact composition) to imitate bone, soft tissue etc. The difference between muscles, fat, bladder etc. And probably add a quantitative table of all these materials, including their density. Authors need to explain the reason on why each material type was chosen, and how these materials correlate to the human body.

The technique implemented in 3d printing process needs to be discussed also.

Specific comments:

L49: Please cite this article together with 2,3. https://doi.org/10.1109/TRPMS.2018.2876562

L50: change thesis to study.

L51: explain PSQA, here it is the first time that is mentioned.

L55: "But the human body..." --> Nevertheless, the human body...

L58: which --> and

L58: "and also....treatments."  Rephrase this sentence.

L59: Rephrase

L61: Rephrase

L63: What stimulation stands for?

L66: "...before the execution of patient treatment." --> before applying the patient treatment

L77: what "...now a day..." stands for?

Table1: Correct the lines of the table.

L118: Explain with more detail what equivalent means. How is it equivalent and why.

I have include my comments on the English of the article in the previous section. The article needs some moderate edit in the English used in the whole article.

Author Response

Thank you for giving me the opportunity to submit a revised draft of my manuscript titled Development of anthropomorphic heterogeneous pelvic phantom and its comparison with homogeneous phantom in advance radiation therapy: Dosimetry Analysis to MDPI. We are complying your valuables suggestions and comments in the revised manuscript.

Reviewer 3 Report

In this study the authors tried to compare the patient-specific absolute dosimetry using homogeneous slab phantom and anthropomorphic (AHFP) heterogeneous female pelvic phantom in cervical cancer patients. They conclude that patient-specific absolute dosimetry should be performed using a heterogeneous (AHFP) phantom that closely resembles the actual human body in terms of both density and design. The paper faced an interesting argument, however there are several issues that need to be re-evaluated in its current form.

1.The author should discuss the results of HU number (bladder and fat) from AHFP phantom.

2. In this research, please add more discussion about the difference of dosimetry analysis between Slab phantom and AHFP phantom/ planed dose and measured dose. It would be helpful if the authors give example or scenario to support its description. Clarification of this point in text is needed

3. How was the sample size (n=30) determined? 

4. line 213, please check “ thenthe “

5. Figs 1B and 2B need to improved.

Minor editing of English language required

Author Response

(The authors gave the same response as above.)

Reviewer 4 Report

- The abstract must be summarized, revise some Paper for try to prepare abstract with the same size, addition abstract no need conclusion. In general abstract is for explain manuscript with around 100 words

- Include information about purification materials or sentence “used as received”

- Manuscript has some typos, revise carefully and correct them

- Manuscript has one reference from 2021, including references from 2022 and 2023

Author Response

(The authors gave the same response as above.)

Round 2

Reviewer 2 Report

There are still 2 main points that need be improved.

1) The composition of materials used is not clear. What normal wax means in terms of composition? How much polyvinyl chloride? You need to explain in detail the composition of materials, so that anyone can reproduce this, after it is published. Maybe give the exact supplier or label of the material. You need to be very specific on the whole procedure of the construction of this phantom.

2) How is a ballon similar to the human bladder? You need to explain in more detail why each material was used? How it is similar to the human body?

English were not greatly improved from the previous version.

Author Response

(The authors gave the same response as above.)

Round 3

Reviewer 2 Report

I am not fully convinced on the use of these materials for such a phantom. Nevertheless, the results supports the proposal of the authors. It seems to be a nice work. Please explain the reason on why authors used such materials to create such a phantom.

Author Response

hank you for giving me the opportunity to submit a revised draft of my manuscript titled Development of anthropomorphic heterogeneous pelvic phantom and its comparison with homogeneous phantom in advance radiation therapy: Dosimetry Analysis to MDPI. We are complying your valuables suggestions and comments in the revised manuscript.
